# Topological Photonic Media and the Possibility of Toroidal Electromagnetic Wavepackets

**Masaru Onoda**

Graduate School of Engineering Science, Akita University, 1-1 Tegatagakuen-machi, Akita 010-8502, Japan; onoda@gipc.akita-u.ac.jp

**Abstract:** This study aims to present a theoretical investigation of a feasible electromagnetic wavepacket with toroidal-type dual vortices. The paper begins with a discussion on geometric phases and angular momenta of electromagnetic vortices in free space and periodic structures, and introduces topological photonic media with a review on topological phenomena of electron systems in solids, such as quantum Hall systems and topological insulators. Representative simulations demonstrate both the characteristics of electromagnetic vortices in a periodic structure and of exotic boundary modes of a topological photonic crystal, on a Y-shaped waveguide configuration. Those boundary modes stem from photonic helical surface modes, i.e., a photonic analog of electronic helical surface states of topological insulators. Then, we discuss the possibility of toroidal electromagnetic wavepackets via topological photonic media, based on the dynamics of an electronic wavepacket around the boundary of a topological insulator and a correspondence relation between electronic helical surface states and photonic helical surface modes. Finally, after introducing a simple algorithm for the construction of wavepacket solutions to Maxwell's equations with multiple types of vortices, we examine the stability of a toroidal electromagnetic wavepacket against reflection and refraction, and further discuss the transformation laws of its topological properties in the corresponding processes.

**Keywords:** geometric phase; photonic orbital angular momentum; topological insulator; topological photonic crystal

## 1. Introduction

Physical concepts proposed for a system are sometimes applicable to other systems that initially looks considerably different from the original system. Such concepts can be used to predict novel phenomena in the latter system and to explain a mechanism governing the phenomena. Such a mechanism could be conversely applied to the original system and predict similar phenomena in it. Finally, we come to realize their universality. Here we discuss about some interconnection among such concepts and mechanisms, e.g., band theory, geometric phase, Hall effect, topological phase and so on. "Energy bands" and "band gaps" were originally cultivated in the field of condensed matter theory which concerns electron systems in solids, e.g., natural crystals or artificial periodic structures. These concepts were applied to the old research theme [1] on electromagnetic waves in periodic structures composed of different kinds of dielectrics and magnetic materials, consequently establishing the concept of "photonic crystals" [2–4], which plays an important role in the realization and extension of "metamaterials" [5–8]. The concept of "geometric phase" was initially introduced in an electromagnetic system [9], and became clearly recognized in a quantum system with spin degrees-of-freedom (DOF) for electron systems in solids [10]. Interestingly, this

clear-cut recognition was reapplied to an electromagnetic system, i.e., a photon system, and its validity became clear in a photon system than in electron systems [11]. On the contrary, the vortex structure of an electromagnetic wave became widely recognized to closely relate with the orbital angular momentum of photons [12], a view currently being implemented in the optical and quantum information communication technology [13–15]. Moreover, electromagnetic vortices can appear in periodic structures, such as photonic crystals [16]. This suggests a new kind of internal orbital angular momenta of photon in such systems. These internal orbital angular momenta may be interpreted as quasi-spin DOF and potentially cause a variety of geometric phase effects. Specifically, an electromagnetic wavepacket composed from wave modes with such vortices can have orbital angular momentum perpendicular to its propagation direction. This relation between angular momentum and propagation direction for such a wavepacket is similar to that for an atmospheric tornado which shows unexpected exotic motions.

Herein, we theoretically investigate a possible electromagnetic wavepacket with toroidal-type dual vortices, i.e., having a ring vortex inside the wavepacket and a line vortex along its propagation direction. The line vortex resembles that of a Laguerre-Gaussian beam and suggests a finite orbital angular momentum of the wavepacket. This paper is organized as follows. In Section 2, we review the relation between geometric phases and angular momenta, followed by the discussion on electromagnetic vortices in periodic structures in Section 3, which further demonstrates the propagation characteristics of such electromagnetic vortices by conducting numerical simulations on Y-shaped waveguides. In Section 4, we introduce the topological photonic media, while reviewing topological phenomena of electron systems in solids, such as quantum Hall systems and topological insulators. Herein, a class of topological photonic media is interpreted as a photonic version of topological insulator and can be realized as an extension of photonic crystals accompanying the electromagnetic vortices. Moreover, we present another simulation of waveguide propagation via exotic boundary modes of such a medium. In Section 5, referring to the dynamics of an electronic wavepacket around the boundary between a topological insulator and conductor, we consider the possibility of toroidal electromagnetic wavepackets with an argument on a correspondence relation between electronic helical surface states of topological insulator and photonic helical surface modes of topological photonic media. In Section 6, we present an algorithm for constructing wavepacket solutions of Maxwell's equations with multiple types of vortices. Next, we numerically investigate the stability of the toroidal electromagnetic wavepacket in reflection and refraction at interfaces between homogeneous isotropic dielectrics, and reveal the transformation laws of topological charges of line and ring vortices.

In the next section and beyond, we adopt the natural system of units as $\hbar = 1$ ($\hbar$: Dirac constant or reduced Planck constant) and $c = 1$ ($c$: speed of light), unless those symbols are explicitly stated. We will not distinguish between the wavevector $k$ of a plane wave and the momentum $\hbar k$ of a quantum particle derived from second quantization of the wave. Likewise, the frequency $\omega$ of a harmonic wave and the energy $\hbar\omega$ of a corresponding quantum particle will not be distinguished. For convenience on later discussions, we introduce a spherical basis $\{e_k, e_\theta, e_\phi\}$ ($e_k = k/k$) in wavevector space.

## 2. Geometric Phases and Angular Momenta of Electromagnetic Vortices

In this section, we first review the relation that exists between the spin angular momentum and geometric phase of an electromagnetic wavepacket, following those between polarization state and spin angular momentum and between polarization vector and geometric phase. Figure 1 describes the relation between polarization state and angular momentum [17] of a right circularly polarized wavepacket propagating in the *z*-direction (upward in the drawing). The arrows represent the deviation of the wavepacket energy flux density from the product of the energy density and the averaged velocity vector, whereas the hue represents the energy density (cold color < warm color). For simplicity, we considered

the situation wherein the wavepacket spread is sufficiently large with respect to its central wavelength so that its deformation can be ignored. In Figure 1, we can find a clockwise vortex in the view facing the propagation direction (*z*-direction) of the wavepacket, indicating that the right circularly polarized wavepacket has a spin angular momentum in the propagation direction. By contrast, we could not find such structure of energy flux density in a similar plot of a linearly polarized wavepacket in the same scale as Figure 1.

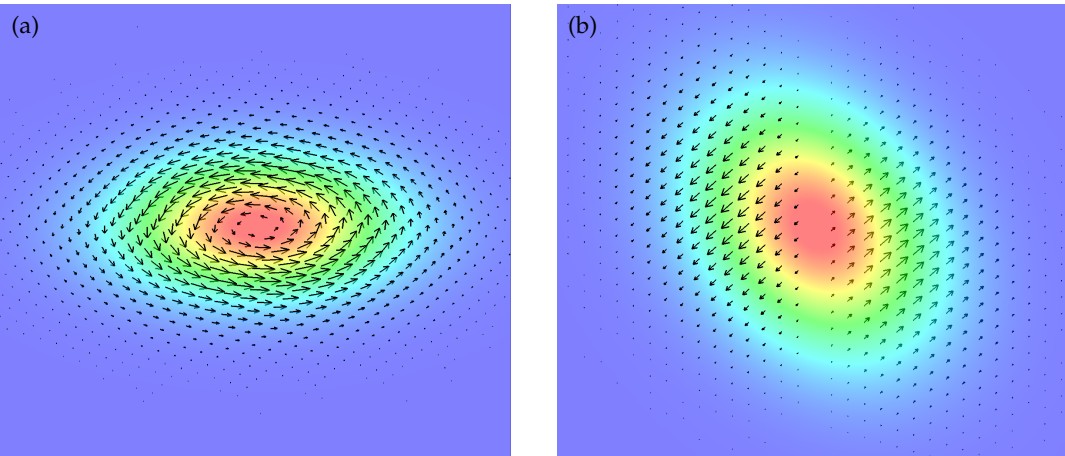

**Figure 1.** (**a**) *xy*- and (**b**) *yz*-cross-sections of a right circularly polarized Gaussian wavepacket.

Furthermore, once the well-known indications, i.e., the geometric phase appearing due to the change in polarization state [9] and the geometric phase due to orbital change of polarized beam [18], are accepted, the relation between spin and geometric phase looms into view. Herein, by specifically looking at the relation between polarization vector and geometric phase, we confirm the relation between the three concepts more explicitly. For that purpose, we introduce two quantities, the Berry connection and curvature, defined as

$$
[\mathbf{\Lambda}_k]_{\alpha\beta} = \begin{pmatrix} -ie_{k\alpha}^\dagger \cdot \frac{\partial}{\partial k^1} e_{k\beta} \\ -ie_{k\alpha}^\dagger \cdot \frac{\partial}{\partial k^2} e_{k\beta} \\ -ie_{k\alpha}^\dagger \cdot \frac{\partial}{\partial k^3} e_{k\beta} \end{pmatrix} = -ie_{k\alpha}^\dagger \cdot \mathbf{\nabla}_k e_{k\beta}, \quad \mathbf{\Omega}_k = \mathbf{\nabla}_k \times \mathbf{\Lambda}_k + i\mathbf{\Lambda}_k \times \mathbf{\Lambda}_k, \tag{1}
$$

where $\{e_{k\alpha}\}$ is an orthonormal basis of polarization vectors normal to the wavevector $k$ ($e_{k\alpha}^\dagger \cdot e_{k\beta} = \delta_{\alpha\beta}$, and $e_{k\alpha}^\dagger \cdot e_k = 0$); the symbol $\alpha$ is an indicator of polarization state; and $e_{k\alpha}^\dagger$ is the complex conjugate transpose (the Hermitian conjugate) of $e_{k\alpha}$. Herein, we also introduce the Jones vector corresponding to this orthonormal basis as $|z_k\rangle$. With $\langle z_k|$ as the Hermitian conjugate of $|z_k\rangle$, they satisfy $\langle z_k|z_k\rangle = \sum_\alpha |z_{k\alpha}|^2 = 1$. Although representations of $\mathbf{\Lambda}_k$ and $\mathbf{\Omega}_k$ depend on how the basis is selected, $\langle z_k|\mathbf{\Omega}_k|z_k\rangle$ is uniquely determined once a state is given. On the basis of right and left circular polarizations, the Berry curvature is represented as $k/k^3 \sigma_3$, where $\sigma_3$ is the third component of Pauli matrices $\sigma = (\sigma_1, \sigma_2, \sigma_3)^\top$ and is diagonal in the standard representation. On the contrary, $\langle z_k|\mathbf{\Omega}_k|z_k\rangle$ can be expressed as $(|z_{kR}|^2 - |z_{kL}|^2)/k^2 e_k$ using the corresponding representation $[z_{kR}, z_{kL}]^\top$ of $|z_k\rangle$. Since the expected value of the spin angular momentum per photon $\langle s_k\rangle$ is evaluated as $\langle s_k\rangle = (|z_{kR}|^2 - |z_{kL}|^2)k/k$, a close relation, $\langle s_k\rangle = k^2 \langle z_k|\mathbf{\Omega}_k|z_k\rangle$, exists between the two quantities.

We can extend the above discussion to a case accompanied by orbital angular momentum. A close relation between the internal orbital angular momentum and Berry curvature is derived in the same way as above, while its discussion gets a little bit complex. Herein, we consider a beam of a central wavevector $k_c$ and introduce the extension

$$e_{k\alpha} \rightarrow \exp\left(il_\alpha \varphi_{k;k_c}\right) e_{k\alpha}, \quad \varphi_{k;k_c} = \arctan \frac{e_{\phi_c} \cdot k}{e_{\theta_c} \cdot k}, \tag{2}$$

where $l_\alpha$ corresponds to the vorticity of the $\alpha$-polarized component and is an integer number, i.e., $l_\alpha \in \mathbb{Z}$. For simplicity, we consider only a class of beams that are a superposition of a given polarization component with the vorticity $l$ and its orthogonal component with $l'$. In other words, we restrict ourselves to a finite subspace of infinite whole space of states. On the basis of right and left circular polarizations, the Berry connection and curvature of this subspace are represented as,

$$\mathbf{\Lambda}_k = -\frac{\cos\theta}{k\sin\theta}(l_{\text{OAM}} + \sigma_3)e_\phi, \quad \mathbf{\Omega}_k = \frac{1}{k^2}(l_{\text{OAM}} + \sigma_3)e_k, \tag{3}$$

where $l_{\text{OAM}}$ is a $2 \times 2$ Hermitian matrix with a pair of integer eigenvalues $(l, l')$. On the other hand, the expected value of the total angular momentum per photon $\langle j_k \rangle$ is evaluated as $\langle j_k \rangle = (z_k | l_{\text{OAM}} + \sigma_3 | z_k) e_k$. We can find a close relation $\langle j_k \rangle = k^2 (z_k | \mathbf{\Omega}_k | z_k)$ again.

Next, we consider the meaning of the form of the Berry curvature, $k/k^3 \sigma_3$, on the basis of circular polarizations which we shall call helicity basis hereafter. As we shall see, this form reflects the photon characteristics of a spin-1 gauge symmetric massless boson. To this end, we introduced a degenerate two-band model of a spin-1/2 fermion system with conical dispersions, similar to relativistic electrons. The Hamiltonian of this model and its projection to the subspace of definite wavevector $k$ are given by

$$H = -iv\nabla_r \cdot \alpha + \Delta \beta \rightarrow H_k = vk \cdot \alpha + \Delta \beta, \tag{4}$$

where $\alpha$ and $\beta$ are Dirac matrices and $v$ and $\Delta$ are parameters of effective phase velocity and band gap, respectively. The band gap of this model plays a similar role as the mass gaps in relativistic theories; hence, we shall refer to the limit $\Delta \rightarrow 0$ as massless limit. The eigenvalue problem of $H_k$ can be easily solved. The eigenvalues of the upper and lower bands are obtained as $\pm\sqrt{v^2 k^2 + \Delta^2}$; only the upper bands will be considered hereafter. The degenerate eigenstates of the upper band $|u_{k\lambda}\rangle$ in terms of the spherical coordinate of wavevector space $(k, \theta, \phi)$ are expressed as

$$|u_{k\pm}\rangle = \frac{1}{\sqrt{2E_k}} \begin{pmatrix} \sqrt{E_k + \Delta}\chi_{k\pm} \\ \pm\sqrt{E_k - \Delta}\chi_{k\pm} \end{pmatrix}, \quad \chi_{k+} = \begin{pmatrix} e^{-i\frac{\phi}{2}}\cos\frac{\theta}{2} \\ e^{i\frac{\phi}{2}}\sin\frac{\theta}{2} \end{pmatrix}, \quad \chi_{k-} = -i\sigma_2 \overline{\chi}_{k+}, \tag{5}$$

where $E_k = \sqrt{v^2 k^2 + \Delta^2}$ and the index $\lambda$ corresponds to the degrees of helicity, i.e., spin component in the direction of $k$. Note that the above expressions are well-defined only in regions excluding the points $\theta = 0$ and $\pi$. However, we shall not step into regularization at these points but only mention that expressions valid at $\theta = 0$ or $\pi$ are obtained by some gauge transformations of the above expressions. One can easily confirm that the above expressions satisfy the time-independent Schrödinger equation in $k$-subspace,

$$E_k |u_{k\lambda}\rangle = \mathcal{H}_k |u_{k\lambda}\rangle. \tag{6}$$

At this point, we would like to mention that the electromagnetic polarization vectors $e_{k\lambda}$ can be interpreted as solutions of a similar Schrödinger-type matrix equation, which is a transcription of Maxwell's equations. The correspondence relation is confirmed by the replacements, $E_k = k/(\sqrt{\epsilon}\sqrt{\mu})$, and

$$\mathcal{H}_k = \begin{pmatrix} 0 & i\epsilon^{-\frac{1}{2}}k \cdot S\mu^{-\frac{1}{2}} \\ -i\mu^{-\frac{1}{2}}k \cdot S\epsilon^{-\frac{1}{2}} & 0 \end{pmatrix}, \quad [S_i]_{jk} = -i\epsilon_{ijk}, \quad |u_{k\lambda}\rangle = \frac{1}{\sqrt{2}} \begin{pmatrix} e_{k\lambda} \\ e_k \times e_{k\lambda} \end{pmatrix}, \tag{7}$$

where $\epsilon$, $\mu$, and $\epsilon_{ijk}$ are the relative permittivity, relative permeability, and the Levi-Civita symbol, respectively. For simplicity, we assumed a homogeneous and isotropic background medium here. Therefore, by means of the state vectors $|u_{k\lambda}\rangle$, we can define the Berry connection in a form common in electronic and electromagnetic systems,

$$[\mathbf{\Lambda}_k]_{\alpha\beta} = -i\langle u_{k\alpha}|\mathbf{\nabla}_k|u_{k\beta}\rangle. \tag{8}$$

Now, let us get back to the degenerate two-band model. On the helicity basis, the Berry connection and curvature of the upper band are expressed by

$$\mathbf{\Lambda}_k = \frac{1}{2k}\left\{\frac{\Delta}{E_k}(-\sigma_2 \mathbf{e}_\theta + \sigma_1 \mathbf{e}_\phi) - \frac{\cos\theta}{\sin\theta}\sigma_3 \mathbf{e}_\phi\right\}, \quad \mathbf{\Omega}_k = \frac{v^2}{2E_k^2}\left\{\frac{\Delta}{E_k}(\sigma_1 \mathbf{e}_\theta + \sigma_2 \mathbf{e}_\phi) + \sigma_3 \mathbf{e}_k\right\}. \tag{9}$$

In the massless limit $\Delta \to 0$, this coincides with the Berry curvature of photons except for the overall coefficient due to a different spin magnitude. In the non-relativistic limit, i.e., an increasing $|\Delta|$ for a fixed $k$, the Berry curvature decreases in the form $1/\Delta^2$, similar to the scale of the spin-orbit interaction. Next, let us consider the relation between the spin angular momentum and the geometric phase in this spin-1/2 massive fermionic system. The spin operator $\mathbf{s}$ and its projection to the upper band $\mathbf{s}_k$ are given as follows:

$$\mathbf{s} = \begin{pmatrix} \boldsymbol{\sigma} & 0 \\ 0 & \boldsymbol{\sigma} \end{pmatrix} \to [\mathbf{s}_k]_{\alpha\beta} = \langle u_{k\alpha}|\mathbf{s}|u_{k\beta}\rangle, \quad \mathbf{s}_k = \frac{1}{2}\left\{\frac{\Delta}{E_k}(\sigma_1 \mathbf{e}_\theta + \sigma_2 \mathbf{e}_\phi) + \sigma_3 \mathbf{e}_k\right\}. \tag{10}$$

In the massless limit $\Delta \to 0$, $\mathbf{s}_k$ matches its photonic version except for the coefficient $1/2$ that comes from the spin-1/2 nature of the present fermionic system. We can again find the simple relation between the Berry curvature and the spin angular momentum, $\mathbf{s}_k = (E_k^2/v^2)\mathbf{\Omega}_k$.

As for the electromagnetic waves in periodic structures, we can develop the same discussion by replacing the polarization vectors with eigenstate vectors expressed using Bloch wave functions, as in the example of a spin-1/2 fermion system discussed above [19,20]. We do not intend to have an unnecessary abstract debate using "Berry connection" and "Berry curvature". Thus, whereas they were introduced in a way that makes the theories under consideration abstract, we can use a common principle that is independent of the details of electron or photon systems for better understanding. Although the definition of angular momentum in a periodic system is accompanied by ambiguity, the Berry curvature can be uniquely defined apart from the freedom-of-choice of basis. Our knowledge of phenomena or effects in a given system are easily applicable to the realization of analogous phenomena or effects in other systems. From this viewpoint, information on the Berry curvature of each band helps us organize the relation between photonic bands in wavevector space and vortices in real space, and serves as guide for controlling vortices.

## 3. Electromagnetic Vortices in Periodic Systems

The spin DOF of photon is no longer well-defined in the presence of periodic structure, while we shall need an alternative concept of photonic spin to discuss the possibility of toroidal electromagnetic wavepackets in Section 5, based on an electron dynamics around the interface of a topological insulator where the concept of electronic spin still works well. (The photonic version of topological insulator will be introduced in Section 4.) One possibility of the alternative is the internal rotational motion of electromagnetic Bloch modes with vortices. In this section, we step further into electromagnetic vortices in periodic structures such as photonic crystals. After looking back briefly on the relation between the Berry curvature and real-space vortex structure in periodic systems, we present a numerical simulation

on propagation modes around a standing vortex mode in a photonic crystal, indicating the effect of the internal rotation in the propagation process.

Useful functionalities of a photonic crystal, such as light confinement and waveguide, are realized by adjusting energy (frequency) dispersion relations and forbidden bands by the periodic structure and symmetry design of the system. The required design procedures stem from the band theory common to wave phenomena in periodic structures. Moreover, based on the studies of geometric Hall effect in electron systems [21,22], we can also use such design to control the Berry curvature of each band [23,24]. For instance, by employing a two-dimensional (2D) periodic system in the $xy$ direction, we can consider a situation where two bands are immediately prior to touching each other at a point $k_0$ in a wavevector space. Given an approximate description, $f_k \pm \sqrt{v^2|k - k_0|^2 + \Delta^2}$ ($f_k$: a function of wavevector $k$) for the local energy dispersion of each, the $z$-component of the Berry curvature of each band is estimated as $\Omega^z_{k\pm} = \pm v^2 \Delta / \{2(v^2|k - k_0|^2 + \Delta^2)\}^{\frac{3}{2}}$. In other words, we can control the Berry curvature by adjusting the level repulsion $\Delta$. We confirmed this mechanism in a more strict theory exactly treating the periodic structure, as well as the relation among the Berry curvature, the angular momentum [19,20] and the real-space vortex structure [16,25]. The electromagnetic vortex can be also controlled through the adjustment of $\Delta$. Figure 2 shows an example of a set of such periodic structure, band diagram, and electromagnetic vortex. As for the 2D photonic crystals, we considered only photonic modes of two distinct polarizations, i.e., transverse-electric (TE) and transverse-magnetic (TM), which propagate strictly parallel to the plane with a 2D periodicity. Herein, we adopted this definition: TE modes have magnetic fields normal to the plane and electric fields in the plane; conversely, TM modes have electric fields normal to the plane and magnetic fields in the plane.

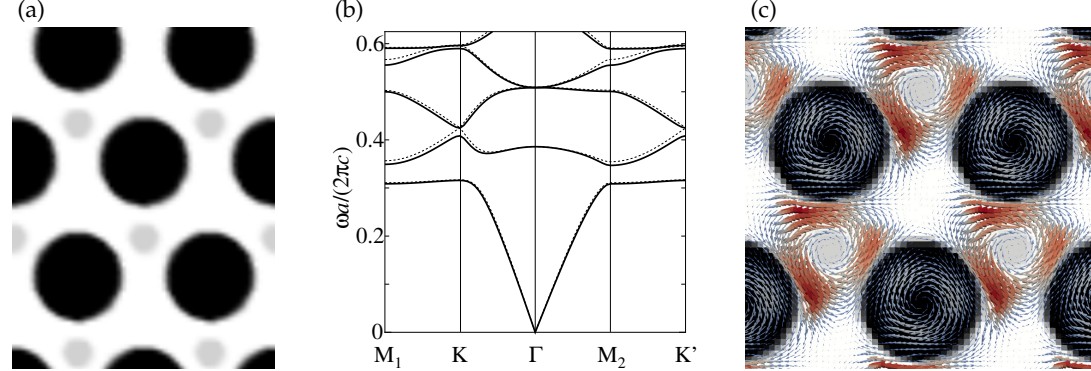

**Figure 2.** (**a**) A sample inversion asymmetric 2D photonic crystal for relative permittivity of 1, 3, and 12 in white, gray, and black regions, respectively. (**b**) Band diagram of TE modes for (**a**). Dotted lines show the case where the relative permittivity of gray rods is set to one. The vertical axis represents the dimensionless frequency $\omega a/(2\pi c)$ (*a*: lattice constant; *c*: speed of light). (**c**) A sample optical tornado: energy flux density of a state at a K-point of the TE 2nd band.

The electromagnetic vortex shown in Figure 2c corresponds to a standing wave mode of a zero group velocity. On the other hand, modes around it have finite group velocities in addition to the vortex structure, and can propagate through the crystal with rotational motion. Next, we describe the propagation characteristics of such modes in a Y-shaped waveguide in Figure 3a composed of the crystal in Figure 2a and a block layer with a sufficiently large band gap covering the relevant frequency range. Since it was not easy to excite a specific electromagnetic vortex mode in a real-time-and-space simulation, we adopted an excitation using a linear source with a line width of a few percent around the central frequency of the targeted vortex modes. Moreover, the linear source was set at the left end of the left branch of the waveguide and the vortex modes were excited by electric field oscillations along the source. Figure 3b

shows the *z*-component of the magnetic field $H_z$ and Figure 3c displays the transmission spectra measured at the ends of the upper right and lower right branches. As the vertical axis is in arbitrary unit, we also plotted the spectrum (blue and red broken lines) of the case where the Y-shaped region is replaced by vacuum, for comparison. Two broken lines overlapped each other, and only the blue broken line is visible. The transmission spectra of the target system in Figure 3c is extremely asymmetric, whereas we find only weak asymmetry in the real-space image of Figure 3b. Figure 3b also shows that this system contains accidental edge modes localized around interfaces aside from bulk vortices; therefore, the asymmetry could not be attributed solely to the bulk vortex modes. An additional simulation (not shown here) confirmed that the edge modes in this system are strongly reflected at the bents of the Y-shaped waveguide; therefore, we conclude that the bulk vortex modes contribute primarily to the asymmetric propagation.

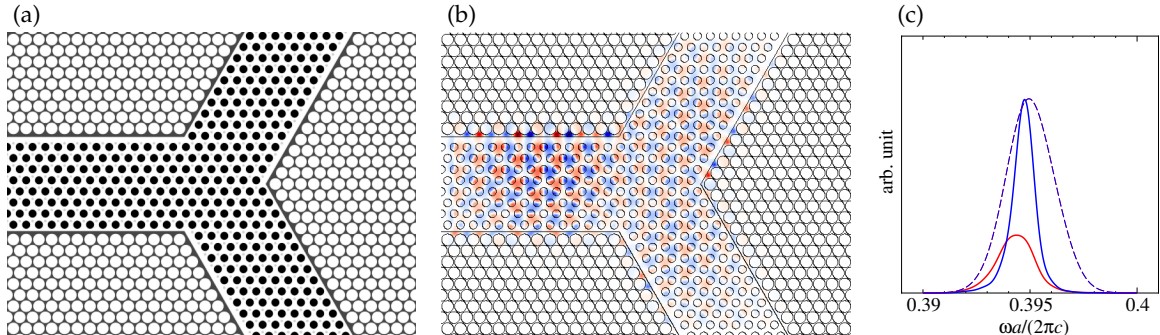

(a)         (b)         (c)

**Figure 3.** (**a**) Y-shaped waveguide composed of the photonic crystal in Figure 2. The relative permittivity of the gray region of block layer around the waveguide is set to be 9. (**b**) *z*-component of magnetic field $H_z$. (**c**) Spectra of the transmissions to the upper right (blue line) and to the lower right (red line) branches. The broken lines represent the transmission spectra for a vacuum Y-shaped region.

## 4. Topological Photonic Media

Based on the pioneering studies on quantum Hall effect in 2D electron systems [26–28], a topological invariant, known as Chern number or index, is assigned to an isolated band via integration of its Berry curvature over the entire first Brillouin zone. Depending on total Chern numbers of bulk bands below a bulk gap, exotic states localize at the edge of a finite system and form edge bands traversing over the bulk gap [29,30]. Furthermore, each state works a one-way waveguide; therefore, is called a chiral edge state. For a Chern number of an isolated band to be nonzero, time-reversal symmetry breaking is necessary. This symmetry may be broken not only by an externally applied magnetic field but also by a spontaneously induced magnetic order [28]. When the quantum Hall effect is induced by the latter mechanism, it is sometimes called the spontaneous quantum Hall effect, as distinguished from the original one. To realize a similar situation in photon systems, the above mechanism requires isolated bands to form with band gaps stemming from time-reversal symmetry breaking. For instance, a magnetic body of a complex permittivity tensor with imaginary off-diagonal components breaks time-reversal symmetry for the photon system. At least the necessary conditions are satisfied by designing a 2D periodic structure made of such a material to form isolated photonic bands. This analogy was the basis for a photonic version of the quantum Hall system theoretically proposed [31–33] and experimentally confirmed [34,35]. Presently, a clear-cut demonstration has also been made on nonreciprocal lasing from chiral edge modes surrounding a network of topological cavities in arbitrary geometry [36].

In reverse, a one-way propagation of a chiral edge mode inevitably breaks the time-reversal symmetry. (Time-reversal symmetry breaking is a necessary condition for the presence of a single chiral edge state; conversely, the presence of a single chiral edge state is a sufficient condition for the symmetry breaking.)

Fortunately, however, chiral edge states and their time-reversal partners can simultaneously exist in a single system preserving time-reversal symmetry in its entirety [37]. As an extension of quantum Hall system to the case with time-reversal symmetry, an insulator with a topologically-protected pair of edge states has been proposed [38]. Such insulator and edge states are respectively called topological insulator and helical edge states [39]. The propagation direction of a helical edge state is selectively governed by its spin polarization. In a naive picture, topological insulator is understood as a superposition of a spontaneous quantum Hall system spin-polarized in a specific direction and its time-reversal partner spin-polarized in the opposite direction which is necessary to maintain time-reversal symmetry. Based on this situation, it was initially called quantum spin Hall system. More precisely, the parity of the numbers of Kramers pairs of helical edge states is critical [38] and corresponds to the topological invariant called $\mathbb{Z}_2$ index. As the index can be calculated by means of the bulk states of a system with periodic boundary conditions, topological insulators can be distinguished from non-topological insulators even with bulk information alone. Furthermore, topological crystalline insulators were proposed by introducing combinations of crystalline symmetries and time-reversal symmetry [40], extending to their photonic version [41–43]. A wider range of topological materials, including superconductors, have been systematically classified based on symmetry and dimensionality [44,45].

Typical physical conditions under which topological insulators emerge are as follows: (1) two pairs of bands with different parities opposite to each pair, which are energetically close to each other, hybridize with each other through a strong spin-orbit interaction and; (2) the resultant level repulsion forms an enough sized bulk gap [46]. By contrast, for photon systems in periodic structures, what kind of DOF should be regarded as spin DOF remains unclear. Nevertheless, if the difference between certain degenerate modes is approximately regarded as pseudo-spin DOF and the coupling between electric and magnetic fields introduced by an artificial chiral medium are regarded as effective spin-orbit interaction of photon, then a similar mechanism could be applied to photon systems in periodic structures. Photonic versions of topological crystalline insulator using metamaterials as artificial chiral media have been proposed [41,42]. After such proposals, it has been pointed out that those topological photonic media could be realized even by photonic crystals composed of only ordinary dielectrics [43], whereas such systems needed to introduce some complication to the unit cell of the crystal. Therefore, we should regard the chiral medium as an example of effective spin-orbit interaction implementation, and not an item of necessity. Figure 4a,b show examples closely related to the all-dielectric topological photonic crystals in Ref. [43]. The structure of Figure 4a is an inversion asymmetric deformation of a topological photonic crystal; hence, we shall call it a quasi-topological photonic crystal for convenience. Compared to the case of Figure 2a, the degree of the symmetry breaking is so weak that it is not easy to distinguish two kinds of rods colored by dark gray and black. The bulk bands are almost unchanged from the symmetric case as depicted in Figure 4c. However, as we shall see below, this symmetry breaking clearly resolves the degeneracy of edge modes and opens a recognizable gap in edge bands. The crystal comes into the topological phase when the inversion symmetry is restored by setting the same values of relative permittivity, i.e., 10, in the dark gray and black regions. Contrastingly, the crystal of Figure 4b is in non-topological phase. Figure 4c is the band diagram of the TM modes of the quasi-topological photonic crystal in Figure 4a. We can find sufficiently-sized band gap at approximately 0.5 in the unit $\omega a/(2\pi c)$. Figure 5a displays the unit cell of the superlattice composed of the crystals of Figure 4a,b. Figure 5b is a closeup of the projected band diagram of TM modes in the superlattice. The edge modes are emphasized in red. The red-dotted lines are the edge modes when the inversion symmetry of the middle part is restored. In this superlattice, the structure around the edge part also breaks the inversion symmetry of the whole system, as evidenced by a small gap in the edge band modes even when the middle part is in the topological phase. The explicit breaking of the inversion symmetry in the middle part increases the size of this gap as well as resolves the degeneracy of the edge modes. Figure 5c gives the energy flux density of an edge mode belonging

to the lowest branch. We can see that the mode is well confined around the boundary and accompanies some eddies.

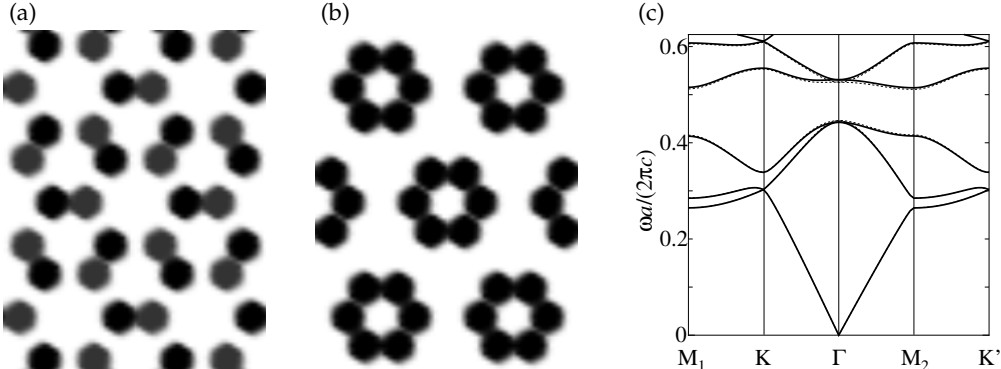

**Figure 4.** (**a**) A sample inversion asymmetric 2D quasi-topological photonic crystal for relative permittivity of 1, 9, and 11 in white, dark gray, and black regions, respectively. When the inversion symmetry is restored, the crystal can be in the topological phase. (**b**) A sample 2D photonic crystal in non-topological phase for relative permittivity of 1 and 10 in white and black regions, respectively. (**c**) Band diagram of TM modes of the photonic crystal in (**a**); dotted lines show the case where the relative permittivities of the gray and black rods are set to 10. The vertical axis represents the dimensionless frequency $\omega a/(2\pi c)$. (*a*: lattice constant and *c*: speed of light).

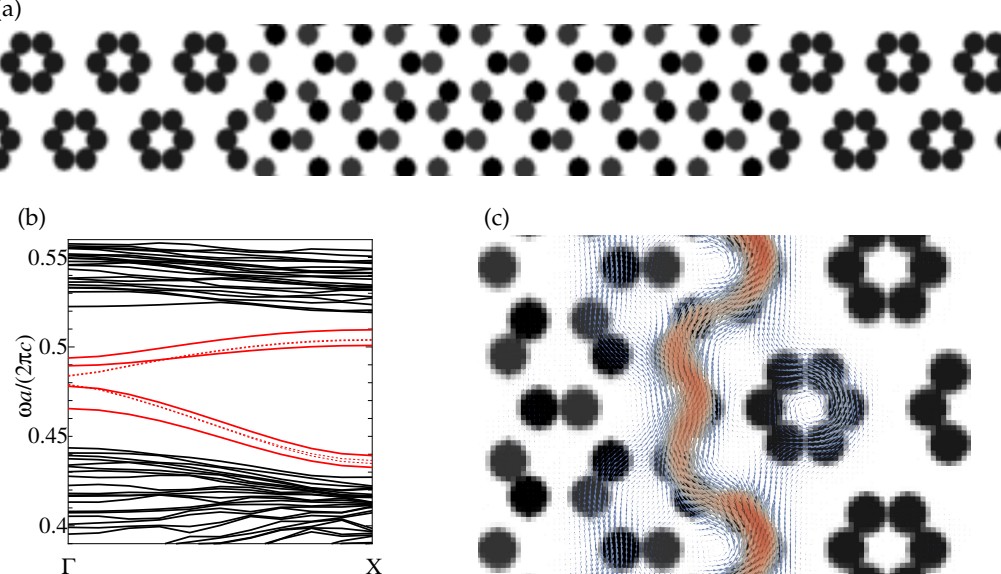

**Figure 5.** (**a**) Unit cell of a 2D superlattice composed of quasi-topological and non-topological photonic crystals in Figure 4. (**b**) Projected band diagram of TM modes of (**a**). Edge modes are emphasized in red. The red-dotted lines show the edge modes where the relative permittivities of the dark gray and black rods in the middle part are set to 10. The vertical axis represents dimensionless frequency $\omega a/(2\pi c)$. (*a*: lattice constant and, *c*: speed of light) (**c**) Energy flux density of an edge mode ($\omega a/(2\pi c) = 0.455$) belonging to the lowest branch (the lowest red curve) in (**b**).

The propagation characteristics of the edge modes were demonstrated in a Y-shaped waveguide in Figure 6a, where the Y-shaped part is composed of the crystal in Figure 4a, whereas the block layer is composed of the crystal in Figure 4b with a sufficiently large band gap to cover the relevant frequency

range. The linear source was set at the left end of the left branch of the waveguide, and the vortex modes were excited by electric field oscillations along the rods, or specifically, in the vertical direction to the page. Figure 6b shows the *z*-component of electric field $E_z$. Here, we can see an extremely anisotropic propagation through helical edge channels. Figure 6c shows the transmission spectra (blue and red solid lines) measured at the ends of the upper and lower right branches along with the spectra (blue and red broken lines) of the case where the Y-shaped region is replaced by vacuum, for comparison. The blue and red broken lines should overlap for the idealistic simulation treating each rod as a material with an exactly sharp boundary. However, smearing each boundary was introduced in the real simulation and the order of introducing the parts influenced the actually simulated structure. For the present case, the simulated structure of the block layer weakly broke the space inversion symmetry; hence, a small discrepancy appeared between the blue and red broken lines. By contrast, a large difference appeared between the blue and red solid lines: the transmission characteristics of the target system were quite asymmetric.

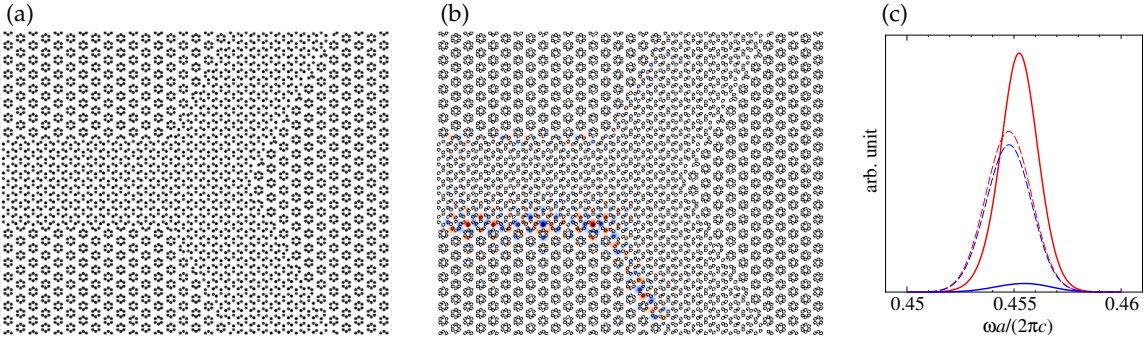

**Figure 6.** (**a**) Y-shaped waveguide composed of quasi-topological and non-topological photonic crystals in Figure 4. (**b**) *z*-component of electric field $E_z$. (**c**) Spectra of the transmissions to the upper right (blue line) and lower right (red line) branches. The broken lines represent the transmission spectra for a vacuum Y-shaped region.

## 5. Electronic State with Twisted Spin-polarization

Focusing on the highly-resolved spin selectivity of helical edge states of topological insulators, we proposed a spin filter using one of the edge states as a conduction channel and a spin control method using hybridization between the edge states and conduction electrons in References [47,48]. For example, we simulated the reflection of an electronic wavepacket at a boundary between the 2D conductor (left side) and topological insulator (right side), as shown in Figure 7, using an effective tight-binding lattice model. For the convenience of numerical treatment, the model was constructed on a simple square lattice, $r = n_1 a_1 + n_2 a_2$, where $n_1$ and $n_2$ are integers and $a_1$ and $a_2$ are primitive lattice vectors of the square lattice. As our focus was on a single-particle state, in principle the first quantization formalism is sufficient. Nevertheless, we introduced the second quantized formalism as a convenient representation method, which enables us to represent operators in compact forms. The Hamiltonian of the conductor part was modeled following a simple tight-binding model,

$$H_{2DC} = \sum_r \left[ \left\{ -t_0 \left( c^\dagger_{r+a_1} c_r + c^\dagger_{r+a_2} c_r \right) + \text{H.c.} \right\} + 4 t_0 c^\dagger_r c_r \right], \tag{11}$$

where $c^\dagger_r$ and $c_r$ are the creation and annihilation operators at a lattice site, and H.c. is Hermitian conjugate. Here, $c_r$ is a spinor operator consisting of up and down spin components, i.e., $c_r = (c_{r\uparrow}, c_{r\downarrow})^\top$. We introduced the last term to adjust the bottom of the conduction band to the origin of energy $E_{k=0} = 0$. Under periodic boundary conditions, the energy dispersion of this model is derived as $E_k = 2t_0 [2 - \cos(k \cdot$

$a_1) - \cos(k \cdot a_2)]$, and which mimics a conventional $k$-square dispersion around the origin of $k$-space, i.e., $E_k \cong k^2/(2m^*)$ $(m^* = 2t_0a^2)$. In modeling the topological insulator, we introduced a spin-dependent $\pi$-flux per square plaquette by the nearest-neighbor hopping of the magnitude $|t_n|$ and classified all the lattice points alternatively to the sub-lattices, A ($n_1 + n_2 \in$ even) and B ($n_1 + n_2 \in$ odd). (The unit cell is doubled, and the primitive vectors of each sub-lattice are given by $a_1 + a_2$ and $-a_1 + a_2$.) Next, we introduced the next-nearest-neighbor hopping of the magnitude $|t_{nn}|$ with alternating signs depending on the sub-lattices and a staggered potential of magnitude $|v_s|$. The Hamiltonian of the topological insulator part is represented by

$$
\begin{aligned}
H_{2DTI} \quad = \quad & \sum_r (-1)^r t_n \left[ \left( c^\dagger_{r+a_1} e^{i\frac{\pi}{4}(-1)^r \sigma_3} c_r - c^\dagger_{r+a_2} e^{-i\frac{\pi}{4}(-1)^r \sigma_3} c_r \right) + \text{H.c.} \right] \\
& + \sum_r (-1)^r t_{nn} \left[ \left( c^\dagger_{r+a_1+a_2} c_r + c^\dagger_{r-a_1+a_2} c_r \right) + \text{H.c.} \right] + \sum_r (-1)^r v_s c^\dagger_r c_r,
\end{aligned}
\tag{12}
$$

where $(-1)^r = (-1)^{n_1+n_2} = \pm 1$ for A and B sub-lattices, respectively. The Hamiltonian $H_{2DTI}$ is time-reversal invariant as a whole as well as $H_{2DC}$, because each of the spin sectors is a time-reversal partner of the other. In the parameter range $4|t_{nn}| > |v_s|$, each of the spin sectors comes in a quantum Hall phase. The spin-resolved quantized Hall conductances have a common absolute value and different signs. They cancel each other so as to preserve time-reversal symmetry. A homogeneously spin-polarized incident wavepacket is illustrated in Figure 7a. The packet is incident from the conductor (left) perpendicularly to the boundary (red vertical line); its spin polarization is uniformly pointing in the incident direction. On the other hand, the spin-polarization state of the reflected wavepacket is depicted in Figure 7b. The spin density of the top white area faces in this side of the page, whereas that of the bottom black area faces the back. Polarization in the vicinity of the wavepacket center has the same state as the pre-incidence. (See Figure 7c for details about the correspondence between spin density and color space.) These results suggest that an electronic state with twisted spin-polarization can be generated from a homogeneously spin-polarized state via topological interface between a conductor and a topological insulator, where helical edge states run along the boundary.

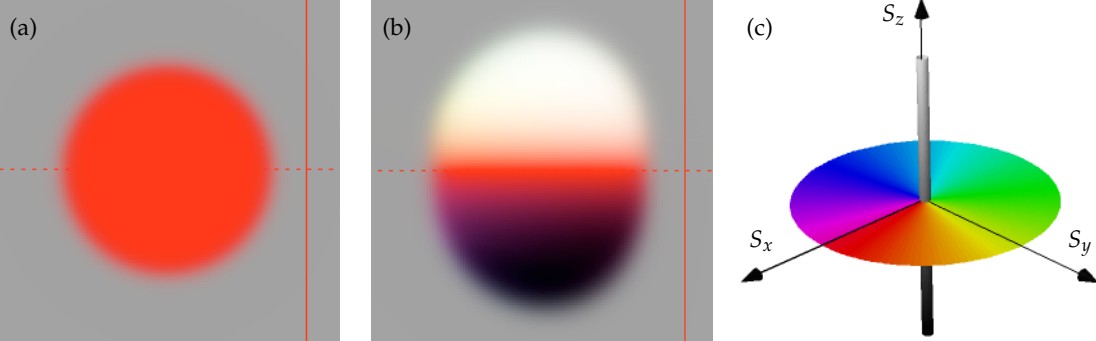

**Figure 7.** (**a**) An incident wavepacket with homogeneous polarization along the $x$-direction and (**b**) a reflected wavepacket with twisted spin texture. (**c**) Relative correspondence between spin density **S** and hue, lightness, and saturation (HLS) color space. Approximately, HLS correspond to the azimuth angle, polar angle, and magnitude of spin density, respectively.

The concept of topological insulator extends to the three-dimensional (3D) system, where helical surface/interface states traversing bulk band gaps emerge [49,50]. Figure 8a shows a conceptual diagram of idealistic helical surface states. The green balls depict the electrons, whose propagation direction and spin angular momentum are indicated by each set of green and black arrows, respectively. For example,

the 2D topological insulator model in Equation (12) can also be extended to 3D versions with some generalizations. A sequence of 3D models is constructed on a simple cubic lattice $\boldsymbol{r} = \sum_\mu n_\mu \boldsymbol{a}_\mu$ ($n_\mu \in \mathbb{Z}$) with an orthogonal set of unit lattice vectors $\boldsymbol{a}_\mu$ ($\mu = 1, 2, 3$). Every site is classified into either A or B sub-lattice as $\boldsymbol{r} \in A(B)$ when $\sum_\mu n_\mu =$ even(odd), in which a sign symbol $(-1)^{\boldsymbol{r}}$ can be introduced as $(-1)^{\boldsymbol{r}} = (-1)^{\sum_\mu n_\mu}$. Moreover, each sub-lattice forms a face-centered cubic lattice. The sequence of 3D models is characterized by three types of parameters $t_\mu$, $t_{\mu\nu}(= t_{\nu\mu})$, and $v_s$, along with SU(2) matrices $\{U_\mu\}$ ($\mu, \nu = 1, 2, 3$) representing the spin-precession processes in the $\mu$-directional nearest-neighbor hoppings. The Hamiltonian of the sequence is given by

$$
\begin{aligned}
\hat{H}_{3DTI} \;=\; & \sum_{\boldsymbol{r}} \sum_{\mu} (-1)^{\boldsymbol{r}} t_\mu \left[ \hat{c}^\dagger_{\boldsymbol{r}+\boldsymbol{a}_\mu} U_\mu^{(-1)^{\boldsymbol{r}}} \hat{c}_{\boldsymbol{r}} + \text{H.c.} \right] \\
& + \sum_{\boldsymbol{r}} \sum_{\mu<\nu} (-1)^{\boldsymbol{r}} t_{\mu\nu} \left[ \left( \hat{c}^\dagger_{\boldsymbol{r}+\boldsymbol{a}_\mu+\boldsymbol{a}_\nu} \hat{c}_{\boldsymbol{r}} + \hat{c}^\dagger_{\boldsymbol{r}-\boldsymbol{a}_\mu+\boldsymbol{a}_\nu} \hat{c}_{\boldsymbol{r}} \right) + \text{H.c.} \right] + \sum_{\boldsymbol{r}} (-1)^{\boldsymbol{r}} v_s \hat{c}^\dagger_{\boldsymbol{r}} \hat{c}_{\boldsymbol{r}}.
\end{aligned}
\tag{13}
$$

This sequence is advantageous in that the edge states of a member with open boundary condition can be analytically investigated in some parameter regions, provided that $\{U_\mu\}$ satisfies the conditions $U_\mu^\dagger U_\nu + U_\nu^\dagger U_\mu = 2\delta_{\mu\nu}\sigma_0$ and $\sigma_2 \overline{U}_\mu \sigma_2 = U_\mu$. Here, $U_\mu^\dagger$ and $\overline{U}_\mu$ are Hermitian and complex conjugates of $U_\mu$, respectively. The symbol $\sigma_0$ stands for the $2 \times 2$ unit matrix in the spin space. Unfortunately, there remain unresolved issues in plausible modeling of the interface between a member of this sequence and conductor. The analysis also accompanies technical complications and will be given elsewhere. Besides, 3D versions of topological photonic crystals have also been proposed [51,52]. Although the relation between a photon's pseudo-spin and actual angular momentum in a periodic structure remains ambiguous currently, examples of energy flux densities of photonic chiral edge modes in References [25,33] and photonic helical edge modes in Figure 5 suggest that the former corresponds to a vortex structure stemming from the latter. A schematic of the ideal photonic helical surface modes is shown in Figure 8b. Here, each set of yellow and black arrows represent the propagation direction and local angular momentum density of a photonic helical surface mode, respectively, whereas the yellow circles containing arrows represent the local vortex structures of the modes.

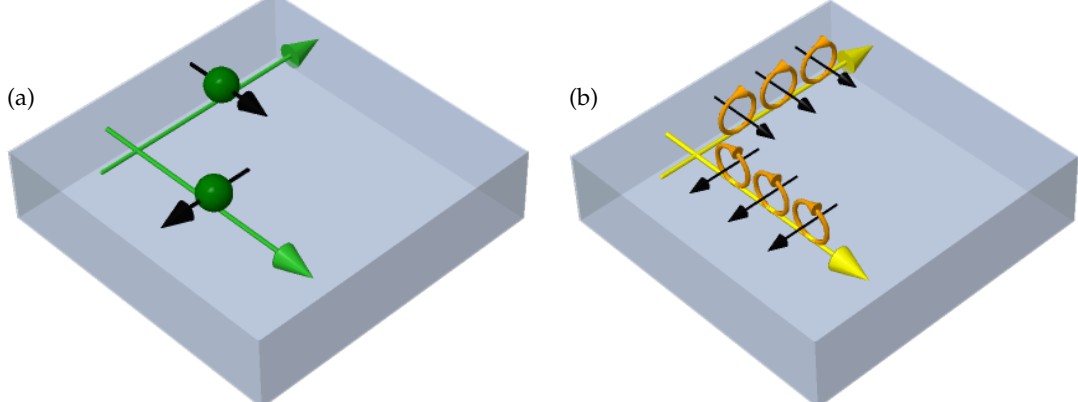

(a)

(b)

**Figure 8.** Conceptual diagrams of (**a**) electronic helical surface states and (**b**) photonic helical surface modes.

Let us look back to the electronic wavepacket with a twisted spin structure in Figure 7b and consider its 3D extension. Suppose a case exists where a wavepacket homogeneously polarized in the propagation direction is perpendicularly incident on the surface of an idealistic 3D topological insulator depicted in Figure 8a. Since in this case we can find the rotational symmetry around the incident axis, the reflected wavepacket is expected to accompany a toroidally-twisted spin texture derived by rotating

Figure 7b around the incident axis. Therefore, the question that arises is how do we extend the above discussion of electoronic wavepacket to its photonic version. Section 4 argues that various types of topological photonic media can be proposed based on the same idea in electron systems. As speculated in Figure 8b, the electronic spin would be replaced by a photonic vortex structure. It is reasonable to replace the incident electronic wavepacket homogeneously spin-polarized to the propagation direction by an electromagnetic wavepacket with photonic orbital angular momentum in the propagation direction, i.e., by that as depicted in Figure 9. Similarly, a simple thinking on the reflected wavepacket would suggest that a toroidally-twisted spin structure can be replaced by a toroidal vortex structure, as depicted in Figure 10. (The hue and the arrows in Figures 9 and 10 represent the energy density and the deviation of energy flux density, respectively, as in Figure 1.) This speculation appears to be extremely naive, because in general, the vortex structure of a photonic helical surface mode is complicated, as displayed in Figure 5c. Nevertheless, as long as the focus is on the topological information of wavepackets, e.g., a set of topological charges of multiple-vortex structure, some realistic vortices are very likely to belong to the same topological class as in Figure 5c, as was the case for Laguerre-Gaussian beams. Therefore, studying the possibility and stability of a photonic/electromagnetic wavepacket with such toroidal vortex structure is worthwhile, not only from an academic point-of-view but also from an application perspective.

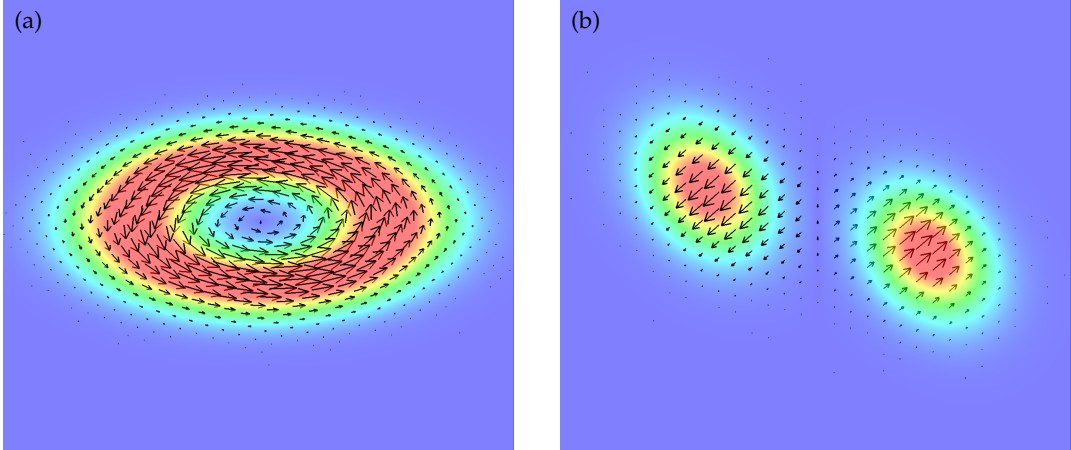

**Figure 9.** (**a**) *xy*- and (**b**) *yz*-cross-sections of a linearly polarized Laguerre-Gaussian wavepacket with orbital angular momentum directed to the positive *z*-axis.

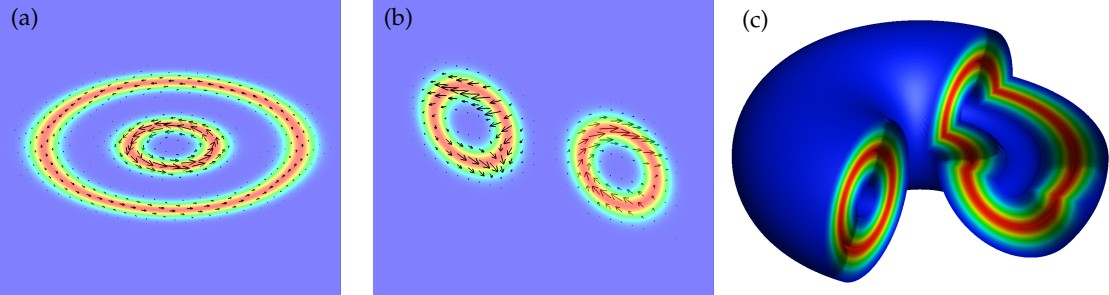

**Figure 10.** (**a**) *xy*- and (**b**) *yz*-cross-sections, and (**c**) isosurface of the energy density of a linearly polarized toroidal wavepacket with orbital angular momentum directed to the positive *z*-axis.

## 6. Propagation Characteristics of Toroidal Electromagnetic Wavepacket

The electromagnetic vortices shown in Figure 9 can be implemented into a quantum digit for information communication [13] and mode division multiplexing in telecommunication technology [14,15].

These applications use the fact that there are multiple quasi-orthogonal modes in a narrow frequency band. Hence, it is a meaningful task to devise various extensions of such electromagnetic vortices. This section brings up the toroidal vortex structure in Figure 10 as one of those extensions. Moreover, it aims to answer questions such as whether the electromagnetic wavepacket with a toroidal vortex can exist as a solution to Maxwell's equations and how stable it is when it can be present.

To answer the initial question, we shall present a procedure to construct wavepacket solutions with generic vortex structures. In the process, we assume that we already have the information of Fourier components $e_{k\alpha}$ of plane wave solutions of mode $\alpha$ with eigenfrequency $\omega_{k\alpha}$, and that the set $\{e_{k\alpha}\}$ ($\alpha = 1, 2, \cdots$) constitutes a perfect orthonormal system, at least at a practical approximation level. The construction procedure consists of four steps as follows:

1.  Construct normalized scalar wavepackets $\{f_\alpha(r)\}$ with trial vortex structures for mode $\alpha$.
2.  Calculate the Fourier transform $\{\tilde{f}_{k\alpha}\}$ of $\{f_\alpha(r)\}$.
3.  Construct the solution of electromagnetic field $\tilde{E}(k, t)$ in $k$-space by

$$\tilde{E}(k, t) = \sum_\alpha \tilde{f}_{k\alpha} z_{k\alpha} e_{k\alpha} \exp\left(i\omega_{k\alpha} t\right),$$ (14)

    where the set of parameters $\{z_{k\alpha}\}$ reduces to Jones vector in simple cases.
4.  Calculate the inverse Fourier transform of $\tilde{E}(k, t)$, and take its real part as the solution $E(r, t)$.

Generally, the above procedure can contain numerical calculations and is inevitably accompanied by approximation due to discretization of both real and wavevector spaces. Nonetheless, in principle the obtained solution converges to an exact solution in the continuous limit. Electromagnetic wavepackets with any vortex structure can actually exist, whereas the stability remains uncertain. In other words, this procedure is applicable as long as the quasi-complete set of $\{e_{k\alpha}\}$ is obtained by either an analytical or numerical method. For instance, a typical case for the former may include the reflection and refraction of linearly polarized plane waves at a flat interface between two different media of homogeneous isotropic permittivity $\epsilon$ and permeability $\mu$. Analytical expressions of $\{e_{k\alpha}\}$ are given by Fresnel's equations, where the index $\alpha$ stands for either P- or S-polarization, while the index $k$ can represent the wavevector of an incident plane wave. As for the latter, we can consider an extension to periodic systems by replacing momentum $k$ by crystal momentum and making mode index $\alpha$ include a band index, along with a degenerate mode index, as in $\alpha \rightarrow n\lambda$ ($n$: band index and $\lambda$: degenerate mode index). Finally, note that the center of a wavepacket can be easily shifted by $r_0$ through the replacement $\{f_\alpha(r)\} \rightarrow \{f_\alpha(r - r_0)\}$.

To simplify the discussion, we shall omit the mode dependence of trial functions introduced above, and pick up only linearly-polarized wavepackets here. Figure 10 provides a sample electromagnetic wavepacket constructed from the procedure above, and which propagates at a positive $z$-direction. The wavepacket has an energy density distributed in a hollow torus-shape, and the wavepacket has a ring-shaped vortex along the internal hollow part of the torus, in addition to a line-shaped vortex associated with the orbital angular momentum directed to positive $z$-direction, which penetrates through the central hole of the torus. Figure 10a represents the $xy$-cross-section of the wavepacket, where we can find the eddy structure of energy flux density corresponding to the orbital angular momentum. On the other hand, Figure 10b represents the $yz$-cross-section of the wavepacket, where we can find another vortex structure whose core corresponds to the hollow part inside the torus. Let us take a closer look at a trial scalar wavepacket with toroidal-type vortex structure, $f_{\mathrm{TWP}}(r)$. This function contains seven types of parameters, namely $m_{\mathrm{line}}$: vorticity of line vortex; $m_{\mathrm{ring}}$: vorticity of ring vortex; $k_{\mathrm{c}}$: central wavevector; $\ell_{\mathrm{R}}$: radius of the central ring inside the hollow region; $\ell_{\mathrm{r}}$: radius of torus-type tube; $\ell_{\Delta}$: thickness of surface layer of hollow torus and; $\ell_{\mathrm{v}}$: size of vortex core, where we set the core sizes of two kinds of vortices to be

the same. By introducing a right-handed basis set $\{e_1, e_2, e_3\}$ with the condition $e_3 = k_c/|k_c|$, we can give an example such as

$$
\begin{aligned}
f_{\text{TWP}}(r) &= \mathcal{N} \left( \frac{z_{\text{line}}}{|z_{\text{line}}|} \right)^{m_{\text{line}}} \left( \frac{z_{\text{ring}}}{|z_{\text{ring}}|} \right)^{m_{\text{ring}}} \tanh \left( \frac{|z_{\text{line}}|}{\ell_{\text{v}}} \right) \tanh \left( \frac{|z_{\text{ring}}|}{\ell_{\text{v}}} \right) \\
&\quad \times \exp \left\{ -\frac{1}{2\ell_{\Delta}^2} \left( |z_{\text{ring}}| - \ell_r \right)^2 + i k_c \cdot r \right\},
\end{aligned}
\tag{15}
$$

$$
z_{\text{line}} = (e_1 + i e_2) \cdot r, \quad z_{\text{ring}} = (|z_{\text{line}}| - \ell_R) + i e_3 \cdot r,
\tag{16}
$$

where $\mathcal{N}$ is a normalization factor. Figure 10 corresponds to the case where $m_{\text{line}} = 1$, $m_{\text{ring}} = -1$, $\lambda = 2\pi/|k_c| = 0.445\ell_0$, $\ell_R = 6\ell_0$, $\ell_r = 3\ell_0$, $\ell_{\Delta} = \ell_0$, and $\ell_v = 0.25\ell_0$ where $\ell_0$ is a unit of length scale. Herein, we shall consider only this set of parameters for toroidal wavepackets, as our focus is limited on the topological properties of toroidal wavepackets, and does not extend to details of their shape deformations. For better understanding by way of comparison, we present trial scalar functions $f_{\text{GWP}}(r)$ and $f_{\text{LGWP}}(r)$ for Gaussian and Laguerre-Gaussian wavepackets, respectively, defined by

$$
f_{\text{GWP}}(r) = \mathcal{N} \exp \left( -\frac{r^2}{2\ell_R^2} + i k_c \cdot r \right),
\tag{17}
$$

$$
f_{\text{LGWP}}(r) = \mathcal{N} \left( \frac{z_{\text{line}}}{|z_{\text{line}}|} \right)^{m_{\text{line}}} \tanh \left( \frac{|z_{\text{line}}|}{\ell_{\text{v}}} \right) \exp \left( -\frac{|z_{\text{ring}}|^2}{2\ell_r^2} + i k_c \cdot r \right),
\tag{18}
$$

The Gaussian wavepacket in Figure 1 and the Laguerre-Gaussian wavepacket in Figure 9 correspond to the cases with $\ell_R = 6\ell_0$ and with $m_{\text{line}} = 1$, $\ell_R = 6\ell_0$, $\ell_r = 2\ell_0$, $\ell_v = 0.25\ell_0$, respectively. In both cases, the wavelength is set at $\lambda = 2\pi/|k_c| = 0.445\ell_0$.

For the second question, we shall consider the stability of the toroidal wavepacket against reflection and refraction on flat interfaces between different kinds of homogeneous isotropic media. The dielectric constants on the lower and upper sides are represented by the symbols $\epsilon_1$ and $\epsilon_2$, respectively. Figures 11–13 show the time lapses for cases with $\epsilon_2/\epsilon_1 = 0.40, 0.75$, and $2.50$, respectively. The incident angle is set at $45°$. The time is measured in units of $\ell_0 \sqrt{\epsilon_1 \mu_0}$. The dimensionless time $\tau$ of each frame is $\tau = -16, -8, 0, +8, +16$ from left to right. Figure 14 shows the incident-angle dependence for $\epsilon_2/\epsilon_1 = 2.50$. The incident angle $\theta$ of each frame is $\theta = 0°, 15°, 30°, 45°, 60°$ from left to right, and the dimensionless time $\tau$ is $\tau = +16$ in every frame. In all cases, the magnetic permeability is set at $\mu = \mu_0$ everywhere, and only the $xz$-cross-sections are depicted. ($x$- and $z$-axes correspond to horizontal and vertical directions, respectively.) We adopt quasi-P-type configuration for the polarization state of every incident wavepacket. (The mean magnetic field of every incident wavepacket is parallel to the interface and normal to the quasi-incident-plane.) From the result in Figure 11, we can presume that the toroidal vortex is stable against reflection. On the other hand, two cases for refraction emerge. First, as the refractive index at the transmission side (Figure 12) decreases, the wavepacket shape stretches in a similar manner to its central wavelength, leading to an unstable ring vortex. Second, as the refractive index at the transmission side (Figure 13) increases, the wavepacket compresses in a similar manner to its central wavelength, resulting in a stable ring vortex at least up to $\theta = 60°$ (Figure 14).

Finally, we would like to mention the transformation laws of the wavepacket topological properties. The vorticity $m_{\text{line}}$ of the linear vortex corresponding to the orbital angular momentum changes as $m_{\text{line}} \rightarrow -m_{\text{line}}$ in reflection, while it does not in refraction, suggested by our analogy with the Laguerre-Gaussian beam with an orbital angular momentum. By contrast, the vorticity $m_{\text{ring}}$ of the ring vortex does not change in both reflection and refraction. In general, recognizing a wavepacket as a particle-like object may give

odd results at first glance. However, the wavepacket is actually a wave phenomenon, and it transforms to get turned inside out in reflection. Since both the rotational flow stemming from the ring vortex and the propagation direction change, the vorticity $m_{\mathrm{ring}}$ defined based on the propagation direction remains unchanged. We would like to conclude this section with a note. As for the cases of partial reflection in Figures 12 and 13, it is not easy to identify reflected wavepackets in the present color contrast due to their weak intensities. Vague reflected wavepackets should appear after extremely increasing the contrast of these figures. On the other hand, in the incident-angle dependence of Figure 14, it becomes possible to recognize reflected wave packets as the incident angle gets away from Brewster's angle for $\epsilon_2/\epsilon_1 = 2.50$ ($\sim57.7°$).

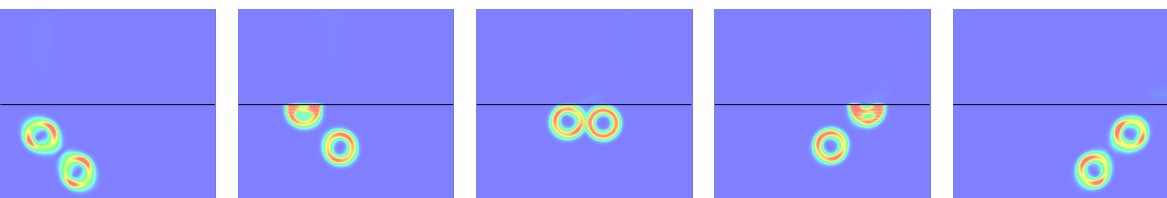

**Figure 11.** Reflection of a toroidal-vortex at the interface of $\epsilon_2/\epsilon_1 = 0.40$ with the incident angle of $45°$.

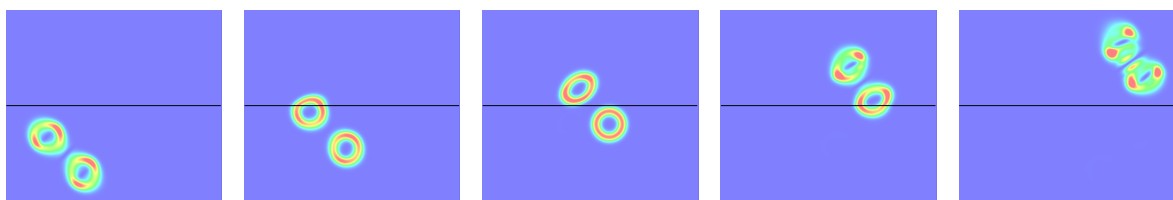

**Figure 12.** Refraction of a toroidal-vortex at the interface of $\epsilon_2/\epsilon_1 = 0.75$ with the incident angle of $45°$.

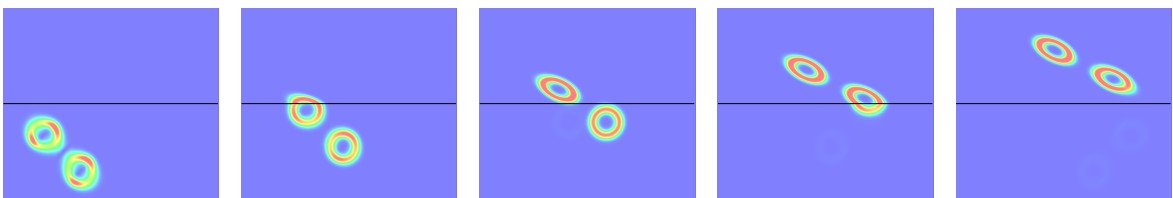

**Figure 13.** Refraction of a toroidal-vortex at the interface of $\epsilon_2/\epsilon_1 = 2.50$ with the incident angle of $45°$.



**Figure 14.** Incident-angle dependence of refracted and reflected toroidal-vortices for $\epsilon_2/\epsilon_1 = 2.50$. The incident angle is (**a**) $0°$, (**b**) $15°$, (**c**) $30°$, (**d**) $45°$ and (**e**) $60°$ from left to right.

## 7. Discussion

Inspired by electromagnetic vortices in free space and periodic structures, and by exotic boundary modes of topological photonic media, we theoretically investigated the topological characteristics and feasibility of a toroidal electromagnetic wavepacket. Our proposal was also based on the numerical analysis of an electronic wavepacket with toroidally-twisted spin structure, generated by a reflection at the interface between an electronic topological insulator and a conductor. We recognized a class of topological photonic

media as the photonic version of the electronic topological insulator and further interpreted it as an extension of a class of photonic crystals where various types of electromagnetic vortex modes emerge with their time-reversal partners. Furthermore, we referred to the fact that modern information transmission technology via electromagnetic waves started to pay attention to photonic orbital angular momentum in a unique way. For instance, optically-based communication technologies have demonstrated the use of photonic orbital angular momentum in the realization of quantum digits for single-photon communication and in the development of a new scheme for multiplexing signals in telecommunications. A key concept common in both examples is the presence of multiple nearly-orthogonal modes within a narrow range of frequencies. From this point of view, we stressed the meaningful benefits of investigating the extensions of this concept, and proposed the toroidal electromagnetic wavepacket as a fusional application with the exotic surface modes of topological photonic media. The electromagnetic wavepacket with toroidal-type dual vortices is an extension of the Laguerre-Gaussian wavepacket whose line vortex corresponds to the photonic orbital angular momentum. Herein, we presented the procedure to construct the solutions of Maxwell's equations with multiple types of vortices. Afterward, we numerically examined the stability of the toroidal electromagnetic wavepacket against reflection and refraction at flat interfaces between the homogeneous isotropic media. Finally, we derived the transformation laws of topological charges of line and ring vortices in these processes.

**Funding:** This work was supported by JSPS KAKENHI Grant Number JP16K05467.

**Acknowledgments:** Band calculations and time-domain simulations for photon systems in periodic structures were performed with free and open-source software packages, MPB [53] and MEEP [54], respectively.

**Conflicts of Interest:** The author declares no conflict of interest. The funders had no role in the design of the study; in the collection, analyses, or interpretation of data; in the writing of the manuscript, or in the decision to publish the results.

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
