# Peer review of "Topological Photonic Media and the Possibility of Toroidal Electromagnetic Wavepackets"

_applsci, doi:10.3390/app9071468_

Round 1

Reviewer 1 Report

This work is very interesting and original study of topological photonic crystals and on propagation and topological protection of toroidal wavepacket in such medium. I find this work very well written and reporting novel interesting science. I recommend publication of the manuscript in its present form.

Author Response

We appreciate the reviewer for spending time on our paper and for the good evaluation of it. We also appreciate the reviewer for recommending the publication of the manuscript in its present form, but we have improved the manuscript following the question and advice of the second and third reviewers. In the process, some of the existing figures and English expressions have also been corrected or improved, while those changes are not critical to the main content and conclusion of the paper. All the changes in the text are highlighted in red. The changes in figures are listed at the end of the reply to the second reviewer.

Reviewer 2 Report

The author has proposed an interesting and important wave packet construction that has a novel vortex structure. The extra angular momentum given to this construction compared to conventional LG beams with only angular momentums along the propagation direction could potentially enlarge the extent of information encoded. This is really nice work, well presented as well. I strongly recommend the publication of this manuscript. 

My only question to the author is, in section 6, could the author comment on the refraction, reflections of these vortex beams according to incident angle? Apart from the 45-degree incident, does the suppressed transmission/reflection happen for other incident angles as well? 

Author Response

We appreciate the reviewer for giving a high evaluation on this paper and for providing a valuable question on the incident angle dependence of refraction and reflection of the vortex beams. As a reply to the question, we have added as Fig.14 the incident angle dependence for the case with the relative permittivity 2.5. Fortunately, we found this result also useful for the supplementary explanation at the end of Sec.6 on the visibility of reflected wavepackets. Other critical improvements have been made following the third reviewer’s advice, and those are explained in the reply to the third reviewer. Some of the existing figures and English expressions have also been corrected or improved, while those changes are not critical to the main content and conclusion of the paper. All the changes in the text are highlighted in red.  The changes in figures are listed at the end of this reply. Finally, we deeply appreciate the reviewer's strong recommendation for the publication of the paper.

Reviewer 3 Report

Review: Topological Photonic Media and the Possibility of Toroidal Electromagnetic Wavepackets

This paper introduced that the electronic state with twisted spin can be generated by a homogeneous spin state going through a topological interface. The author then proposed inhomogeneous or toroidal polarized wave packet can be obtained by a similar process. The manuscript also included a review of topological photonics, describing different types of topological effects and simulation results of topological/trivial Y-shape waveguide.

I find many strengths in the manuscript:

•           The theoretical investigation and the numerical analysis are comprehensive and carried out at the highest level.

•           The manuscript is well-written, and it is easy to follow.

•           The calculation seems complicated and the comparison between electronic system and photonic system of toroidal polarized light have a great potential for many applications.

The subject Toroidal Electromagnetic Wavepackets are drawing great interest and generally appeals to a broad readership.

The topic of this work is suitable for applied sciences. However, there exist several critical issues: on the scientific correctness and the ambiguity. Therefore, at this stage, I suggest the major revision of the manuscript.

1.The connection between section 3-4 and 5-7 is very weak.  I suggest the reviewer present more details and explanations. I also don't think it is necessary to include a review of topological photonics here.  The author should also justify this section.

2. I didn't find the discussion about the possibility to implement such toroidal polarized wave packet. I suggest the author give more explanations rather than focusing on the review of topological photonics.

3. Fig. 4: it is extremely difficult to distinguish grey to black in the figure, I suggest the reviewer change the color of the figure

4. line 263 typo, should be fig. 5

5.For the references the reviewer should cite more relevant papers, ‘’ This symmetry may be broken either artificially by applied magnetic fields or spontaneously by some magnetic order [28]’’

The author should cite the first demonstration of the topological laser using magnetic material to break the time-reversal symmetry (“Nonreciprocal lasing in topological cavities of arbitrary geometries”, Science 358, 636-640 (2017))

Author Response

We appreciate the reviewer for the careful reading of our paper and for providing valuable advice to improve it. We have modified or reexamined the following points recommended by the reviewer and tried to improve the paper.

>>1.The connection between section 3­4 and 5­7 is very weak. I suggest the reviewer present more details and explanations. I also don't think it is necessary to include a review of topological photonics here. The author should also justify this section.

>>2. I didn't find the discussion about the possibility to implement such toroidal polarized wave packet. I suggest the author give more explanations rather than focusing on the review of topological photonics.

Thanks to these suggestions, we have again recognized the importance of various readers' point of view. Our way of presentation on the connection was rough, and the explanation about it might not been enough or clear. We have added the explanation of the connection among Secs.3, 4 and 5 at the beginning of Sec.3. (We think that the connection between Secs.5 and 6 is evident and also the case Sec.7.) Along with this, we have also modified related parts, e.g. the abstract, Sec.1 and so on. Those corrections are also indicated in red as well as all other ones.

We guess that the reviewer would be an excellent expert on topological photonics and also have a high level of insight on other physics. Our way of thinking and discussion in the present study might look roundabout. However, the present style of the paper is necessary for us to organize our thoughts and to explain how we come up to the possibility of electromagnetic toroidal wavepacket from the electron dynamics in topological spin electronics with arguing on the quasi-spin degrees of freedom of photon in periodic structures. The review part might be boring for experts at the reviewer's level. However, we strongly believe that it would be helpful for average researchers like us and beginners about topological spin electronics and photonics. We guess that both types are potential readers of this open-access journal.

>>3. Fig. 4: it is extremely difficult to distinguish grey to black in the figure, I suggest the reviewer change the color of the figure.

We deeply appreciate the reviewer for paying attention even to such a point. Although we also recognize the intent of the suggestion well, we beg the reviewer’s tolerant understanding on our choice not to change Fig.4 but to give an additional explanation about this point in the main text. Before the first submission, we have repeatedly examined this point and have spent a long time adjusting the appearance of this figure and simultaneously those of related ones. We think the appearance of Fig.4 correctly represents the weakness of the inversion symmetry breaking of the corresponding crystal compared to the one depicted by Fig.2, and would like to leave this point. The weakness of symmetry breaking also corresponds to the smallness of the changes in bulk bands from the symmetric case. 

>>4. line 263 typo, should be fig. 5

This typo and other similar errors have been fixed. We appreciate the reviewer's careful attention again.

>>5.For the references the reviewer should cite more relevant papers, ‘’ This symmetry may be broken either artificially by applied magnetic fields or spontaneously by some magnetic order [28]’’

The author should cite the first demonstration of the topological laser using magnetic material to break the time­reversal symmetry (“Nonreciprocal lasing in topological cavities of arbitrary geometries”, Science 358, 636­640 (2017))

We thank the reviewer for introducing the interesting paper. We have mentioned and cited this clear-cut demonstration on topological photonics at the end of the paragraph including the sentence "This symmetry .... order [28]".

In addition to the above critical corrections, as a reply to the second reviewer’s question, we have added in Fig.14 the incident angle dependence for the case with the relative permittivity 2.5. Some of the existing figures and English expressions have also been corrected or improved, while those changes are not critical to the main content and conclusion of the paper. All the changes in the text are highlighted in red. The changes in figures are listed at the end of the reply to the second reviewer. 

Finally, thanks to the reviewer again for the valuable advice for improvement.

Round 2

Reviewer 3 Report

I am satisfied with the author's response to my comments and recommend accepting the manuscript for publication.